# Municipal officials' subjective distress in coordinating with the national government during the decontamination project of radioactive materials in Fukushima: A qualitative study

Tomoo Hidaka[1]*, Takeyasu Kakamu[1], Hideaki Kasuga[2], Yuko Suzuki[3], Toshihiro Terui[4], Shizuka Kawamoto[5], Tatsuya Sato[6]

1 Department of Hygiene and Preventive Medicine, School of Medicine, Fukushima Medical University, Fukushima, Fukushima, Japan, 2 Department of Psychology, School of Psychology, Tokai Gakuen University, Miyoshi, Aichi, Japan, 3 Department of Nursing, Faculty of Nursing, Seitoku University, Matsudo, Chiba, Japan, 4 Department of Neuropsychiatry, School of Medicine, Fukushima Medical University, Fukushima, Fukushima, Japan, 5 General Education Division, Kyoto Seika University, Kyoto, Kyoto, Japan, 6 College of Comprehensive Psychology, Ritsumeikan University, Ibaraki, Osaka, Japan

* thidaka@fmu.ac.jp

## Abstract

Municipal government officials (MGOs) have played key roles in managing decontamination project of radioactive materials in the aftermath of the Fukushima Daiichi Nuclear Power Plant (FDNPP) accident. However, the subjective distress experienced by MGOs during the decontamination project, mainly stemming from challenges in coordination with the national government, is not yet fully documented. The purpose of this study is to descriptively understand the nature and causes of subjective distress among MGOs and to make an exploratory conceptual model of the process through which such subjective distress becomes manifest. Narratives from three MGOs were collected through interviews and subjected to qualitative analysis using the Steps for Coding and Theorisation method. For the first purpose, the results indicate that inconsistencies in national reconstruction and decontamination designs impose substantial burdens at the operational level; these burdens are further amplified by inadequate management by national authorities, and subjective distress emerges through a breakdown of trust among national government officials, municipalities, and residents. For the second purpose, an exploratory conceptual model explaining the progression toward subjective distress through the following four stages was created: the emergence of distrust and adversarial orientations toward the national government, the dilemma of occupying an intermediate position, the erosion of ties with residents, and the threat to professional pride. Overall, through experiences such as questioning the national government's reconstruction policies, enduring the psychological burden of occupying an intermediate position, and witnessing a breakdown in trust with residents, MGOs may find their professional pride undermined; consequently, subjective distress, manifesting as a profound

**Data availability statement:** All relevant data are within the paper and its Supporting information file.

**Funding:** TH received the grant from Japan Society for the Promotion of Science for this study (JSPS KAKENHI, grant number 24K06571). The JSPS had no role in study design, data collection and analysis, decision to publish, or preparation of the manuscript. https://www.jsps.go.jp/english/.

**Competing interests:** The authors have declared that no competing interests exist.

sense of emptiness, may arise. As MGOs are likely to stand at the frontline of managing post-radiation-disaster reconstruction and decontamination efforts, it is essential to develop protective measures for their mental health from both macro-level, organisational perspectives and micro-level, occupational and psychological perspectives.

## Introduction

In Japan, municipal government officials (MGOs) play key roles in long-term disaster recovery [1,2], as seen after the 2011 Fukushima Daiichi Nuclear Power Plant (FDNPP) accident, when they were tasked with managing decontamination project of radioactive materials by the national government [3]. The decontamination project was a large-scale program aimed at removing radioactive materials released into the environment following the FDNPP accident, and it was an essential process for enabling residents to return and for promoting regional reconstruction [3]. Under the government's leadership, municipalities affected by contamination were legally obligated to engage in decontamination work [3]. Therefore, the MGOs were required to assume new tasks and responsibilities, including liaising and coordinating with the national government, which they had not experienced previously.

Under Japan's administrative system, MGOs often operate with limited autonomy, because the national government controls much of the budget and policy direction [4,5]. This structural imbalance complicates the MGOs' ability to adapt national-level decisions to local needs, particularly in unprecedented situations such as post-nuclear disaster recovery. Among the MGOs involved in the recovery process, the introduction of new responsibilities, such as risk communication with residents under national policies and guidelines, and the need to work under extremely harsh conditions, such as sleep deprivation, contributed to widespread emotional exhaustion, depression, and even suicide [6–10]. These mental health problems stem from strained national–municipal relations and the suffering of MGOs caught between conflicting expectations and institutional pressures.

However, little is known about how MGOs subjectively experienced and interpreted such strained relationships in the context of post-FDNPP accident reconstruction efforts. In particular, it remains unclear which specific aspects of national government policies or officials' behaviour MGOs perceived as contributing to their subjective distress. While administrative indicators such as resignations or leaves of absence may signal occupational strain, they do not illuminate how that distress was experienced or meaningfully linked to intergovernmental interactions. An in-depth examination of this intergovernmental interface from the viewpoint of MGOs is therefore essential for understanding how policy implementation processes shape occupational experiences, institutional trust, and barriers to reconstruction following a nuclear disaster.

Exploring the subjective meaning-making of MGOs' experiences is best achieved through analysing their own narratives, which offer deeper insight into how individuals interpret and assign meaning to their experiences within complex institutional contexts. On the other hand, the views of affected MGOs toward the national government

are rarely made public. This may be related to the tendency of Japanese public servants to respect harmony [11] and legal obligations of confidentiality and political neutrality [1]. These characteristics may be the reasons for the reluctance of the affected MGOs to disclose their views regarding the national government, especially when those views are critical. Therefore, when publishing the narratives of MGOs, their anonymity must be protected. In some cases, it is recognised that immediate public disclosure of such narratives may be inappropriate. Accordingly, publication may be delayed until the MGOs themselves judge that disclosure is acceptable, such as after transfers or retirements that reduce the risk of identification. Through such publication strategies, both the protection of MGOs' anonymity and respect for their autonomy in research participation can be ensured.

The purpose of this study is (1) to descriptively understand the nature and causes of subjective distress among MGOs and (2) to make an exploratory conceptual model of the process through which such subjective distress becomes manifest.

## Methods

### Theoretical background of inquiry

This study adopts a constructivist-interpretivist epistemology, assuming reality is socially constructed and meaning arises through individuals' experiences and interpretations of the world around them [12]. We examined how MGOs in Fukushima subjectively understood their roles, responsibilities, and emotional burdens during radioactive material decontamination, particularly in relation to their interactions with national government actors. The analysis focuses on the narratives and meanings MGOs assigned to their institutional relationships and work contexts, rather than on objective "truths." This epistemological approach informed the study's design, data collection, and analysis.

### Study design and recruitment of participants

In this cross-sectional qualitative study, semi-structured interviews were conducted with three participants: Mr. X, Mr. Y and Mr. Z. They were full-time permanent employees of one of the 12 Fukushima municipalities that were designated as evacuation zones after the FDNPP accident (Tamura, Minamisoma, Kawamata, Hirono, Naraha, Tomioka, Kawauchi, Okuma, Futaba, Namie, Katsurao and Iitate). Table 1 shows participant profiles. All participants were born and raised in the municipality in which they were working, and were therefore familiar with the geographical, industrial, and demographic conditions of the area. Several aspects of their backgrounds are particularly relevant to the present study: (1) they had spent their entire careers in non-specialist administrative positions; (2) they had no professional training or work experience in dealing with radiation hazards prior to the FDNPP accident; (3) they had been dispatched to a government corporation established by their municipal office to oversee and manage decontamination work within their municipality from immediately after the FDNPP accident until the time of the survey; and (4) they were therefore deeply involved in communication and coordination with both the national government and local residents after the decontamination work started. Note that participants' gender was reported to provide contextual information that enables readers to assess the narratives' interpretability, consistent with the principles of thick description in qualitative research. In the study context, all MGOs involved in coordination with the national government were male, and thus reporting gender did not increase the risk of identification or compromise participant anonymity.

**Table 1. Interviewee profiles.**

| Interviewee | Gender | Age | Area of responsibility | Years of service at the municipal government (approx.) |
|---|---|---|---|---|
| X | Male | Late 40s | Construction and Infrastructure | 25 years |
| Y | Male | Early 50s | Agriculture and Livestock promotion | 30 years |
| Z | Male | Early 40s | Agriculture and Livestock promotion | 20 years |

According to criteria for interviewee adequacy in qualitative research, it is essential to include participants who possess sufficient knowledge and experience of the subject matter [13]. The number of interviewees is typically determined by practical factors, such as the depth and duration of interviews as well as the availability of interviewers [14–16]. Accordingly, we selected three municipal officials – Mr X, Mr Y, and Mr Z – who fulfilled the criteria of interviewee adequacy in qualitative research, based on their relevant and reliable knowledge and experience, as shown in the participant profiles [17]. Although the number of interviewees may appear small, we adopted a quality-over-quantity approach, which is commonly employed in qualitative research such as resentment or skepticism toward the national government among MGOs, the limited number of interviewees was justified from both methodological and practical standpoints.

The first and second authors were already acquainted with all participants prior to the start of this study through a health support project for local recovery workers. As part of this project, the authors visited the participants' municipality and developed friendly relationships with them, including daily informal conversations. Through these interactions, the authors and participants developed a mutual understanding of each other's personalities, backgrounds, and areas of expertise/responsibility. All participants provided written informed consent for both participation in the interviews and publication of the study.

## Data collection

The interviews and analyses in this study were conducted in Japanese. The excerpts quoted below were translated into English for the purpose of this paper.

Semi-structured face-to-face interviews were conducted between 2015 and 2017, with the first author serving as the interviewer. The number of sessions and the duration of each session varied according to the participants' schedules. We conducted one interview with Mr X and Mr Y in 2015, and five interviews with Mr Z between 2016 and 2017. The total duration of the interviews was 525 minutes, with an average of approximately 75 minutes per session. All interviews were audio-recorded using an IC recorder with the participants' permission and transcribed verbatim for qualitative analysis.

To explore the nature of subjective distress experienced by the municipal officials in the context of recovery work, the following questions were addressed: (1) what kinds of difficulties or stresses were the participants experiencing in their reconstruction-related work, including decontamination; (2) how they had perceived these challenges over time; and (3) what concerns the participants had about the future of their community and municipal administration. Additional follow-up questions were asked as needed.

In this study, we operationally defined subjective distress not as a specific illness or disorder, but as a broad concept encompassing discomfort, dissatisfaction, and emotional unease, grounded in the interviewees' lived experiences. Importantly, these questions focused on general difficulties rather than directly inquiring about the participants' relationship with the national government. This approach enabled us to explore how their subjective distress stemmed from such relationships and/or other contributing factors.

From 2015 to 2017, radioactive material decontamination work was ongoing, and the process of lifting evacuation orders had begun, leading to the return of residents [18]. Therefore, at the time of the interviews, the interviewees were in the midst of experiencing difficulties in their relationships with both national government officials and local residents regarding decontamination and reconstruction work. They recalled the confusion that had occurred in the immediate aftermath of the 2011 earthquake and resulting nuclear disaster, as well as the subsequent reconstruction process, and were able to provide detailed responses in the interviews.

## Qualitative analysis

The verbatim transcripts of all the interviews were analysed cross-sectionally using Steps for Coding and Theorisation (SCAT), a qualitative data analysis method for inductively summarizing narratives, as described in previous studies

 

[19–21]. SCAT has the advantage of extracting important excerpts from the original narrative and gradually abstracting them, ultimately deriving concepts while considering the meanings embedded in the original narrative. Specifically, we used the following seven steps: (1) extracting narratives that were considered important and rendering them into text; (2) selecting noteworthy words or phrases from the text (Step 1) to set the focus of the analysis; (3) paraphrasing the words and phrases extracted in Step 2 to convert the interviewee's unique expressions into analyzable and interpretable forms; (4) identifying underlying regularities such as cause-and-effect relationships, sequential patterns, and/or tendencies within the data by using concepts from academic or professional domains that account for paraphrased texts in Step 3; (5) describing the core meanings expressed in the original narratives in abstract terms as concise headings—composed of nouns or noun phrases and referred to as "themes"—based on the results of Steps 2–4, in a way that generates novel conceptual constructs; (6) constructing a storyline that explains the interviewee's experiences by using all themes generated in Step 5, supplementing them with additional wording where necessary; (7) by articulating the storyline in a generalized and unified descriptive form, describing the "theory" which is the brief and declarative expression what can be claimed based on the analysis of the text. The theory generated in Step 7 is the outcome of SCAT. Note that SCAT analysis is, in principle, conducted for each interviewee; accordingly, in this study as well, three distinct theories, corresponding to the number of interviewees, were ultimately generated.

As above-mentioned, SCAT allows gradual abstraction from transcribed texts to conceptual categories while maintaining explicit and systematic correspondence between the source text and the generated codes, resulting in a high degree of falsifiability of analytical results [22]. SCAT was considered an appropriate method for the current study, as such methodological advantages had been previously demonstrated in studies on fear of radiation health risks and concerns about returning home after the FDNPP accident among evacuees [23,24].

During the analysis, we focused on speeches in which interviewees described tensions with the national government or criticized its bureaucratic structures, often using emotionally charged language such as anger, sadness, frustration, meaninglessness, and bewilderment. These narratives were viewed as reflections of the MGOs' experiences, and were therefore treated as the analytical target/material in the coding process. The first author conducted the analysis, and the third author verified whether the analysis results accurately reflected what had actually been spoken. Both authors have more than ten years of experience in qualitative psychology research. The analysis was conducted using the SCAT coding form created in Excel 2019, which was provided by the developer of SCAT [25] and, as a result, 38 descriptions of theories were identified: 12 from Mr X, 8 from Mr Y, and 18 from Mr Z. These theories and underlying themes and storylines, as listed on the SCAT coding form, are shown in S1 Table.

In the original SCAT procedure, the analysis is conducted and reported for each interviewee. However, because this study aimed to identify overarching patterns across participants, the researcher conducted an additional interpretive integration process that is not part of the standard SCAT procedure. Specifically, the individual theories generated through SCAT were further examined, compared, and conceptually grouped into higher-order categories through interpretive abstraction. This supplementary step was conducted in the discussion section to enhance integrative understanding while maintaining fidelity to the original SCAT principles by developing an exploratory conceptual model of the process by which MGOs' subjective distress becomes manifest.

### Saturation assessment

Although this study does not employ a Grounded Theory Approach, and therefore does not adhere to the concept of 'theoretical saturation' as defined in that methodology, it is still important to address the notion of 'saturation' in the more general sense of data and analytical sufficiency in qualitative research. In this study, we adopted the 'information power' framework [26], which evaluates sample size and saturation in qualitative research from five key perspectives: study aim (narrow/broad), sample specificity (dense/sparse), use of established theory (used/not used), quality of dialogue (strong/

weak), and analysis strategy (case/cross-case). According to this framework, studies with a narrow aim, a dense (specific) sample, an established theory, high-quality dialogue, and a case-focused analysis strategy are considered to possess greater information power and, therefore, can justify small sample sizes.

In the context of this study, the study aims incorporated both narrow and broad foci: the exploration of subjective distress experienced by MGOs represents the narrow focus, while the development of an exploratory conceptual framework and practical countermeasures reflects the broad focus. However, the scope of the exploratory conceptual framework was limited to the specific context of post-radiation-disaster decontamination operations, rather than generalised to disasters more broadly. Regarding sample specificity, the participants constituted a dense sample of MGOs who had been specifically engaged in coordination and negotiation with the national government during radioactive material decontamination projects. As for the use of an established theory, none was adopted, as the aim of this study was to construct a new theoretical framework to inductively understand the experiences of MGOs during the reconstruction period. Regarding the quality of dialogue, the authors had pre-existing relationships with the interviewees, and in the case of Mr Z, multiple interviews were conducted, allowing for the collection of rich, in-depth narrative data. Lastly, the analysis strategy employed case-level analysis instead of cross-case comparisons. Based on these criteria from the information power framework, a small sample size was deemed appropriate for the present study.

The narratives provided by the interviewees were highly varied and richly detailed, encompassing a wide range of content—from objective information related to laws and institutional conditions to emotional responses such as anger and sorrow. This richness of content was deemed sufficient to yield meaningful insights; thus, additional participant recruitment was considered unnecessary. Based on both the above theoretical assessment and this analytic evidence, we argue that a sample size of three participants was appropriate for this narrative study, and that the findings are supported by sufficient data and analytical depth.

### Ethics

Written informed consent for participation in and publication of this study was obtained from all participants prior to their first interview. The study protocol was approved by the Research Ethics Committee of Fukushima Medical University, Fukushima, Japan (Approval No. 2530).

### Results

The theories generated through the SCAT analysis were described for each interviewee (Table 2). For convenience, theories are listed and numbered according to the order in which they were generated.

### Mr X

From Mr X's narrative, twelve theories were generated. The theories visualised the structure in which inconsistencies in national reconstruction/decontamination design produce burdens at the operational level, ultimately resulting in the exhaustion of individual MGOs.

Theories X2–4 showed that in a decontamination project required after a nuclear power plant accident, MGOs are not necessarily ready to begin their work. These theories indicate that MGOs may be compelled to engage in the project despite inadequate preparation, and that national support to compensate for such unpreparedness may also be insufficient. The following narrative excerpt constitutes one of the elements of theory X3:

> (In response to the question: 'Did you have time to prepare carefully for your involvement in the decontamination and reconstruction project?')

> "There was no time. We had experience in ordering construction work in our administrative job, so we had the know-how to order and request construction companies and transport companies involved in the radiation decontamination

**Table 2. Theories from Interviewees' narratives.**

*Mr. X*

1. Discrepancies between practical realities and theoretical models of reconstruction become evident soon after the initiation of post-disaster recovery efforts.

2. While national frameworks envision a linear, well-coordinated process, municipalities on the ground confront fragmented responsibilities, unpredictable challenges, and significant institutional gaps.

3. Municipal government officials' unexpected participation in radiation decontamination projects creates a mismatch: a municipality that was not originally positioned to lead such complex technical operations suddenly finds itself at the centre of implementation.

4. The municipal officials face limited time, autonomy, and decision-making authority, and are forced to devise local responses in the absence of national governmental support.

5. A growing dissatisfaction among municipal officials with national radiation exposure standards is evident, with many regarding them as disconnected from residents' concerns and local realities.

6. Municipal officials bear excessive responsibility for planning and timing the lifting of residents' evacuation orders and support, despite the immense social, ethical, and practical implications of such decisions.

7. Officials are responsible for reporting to the public on the results and limitations of decontamination, serving as intermediaries between national policy and community expectations.

8. The municipal officials face serious operational issues, including dealing with workers exhibiting poor behaviour, logistical challenges, and the emotional weight of supporting residents in returning to and living in their hometowns with a sense of safety, even when uncertainties remain.

9. The misalignment of ultimate decontamination goals between national and municipal governments exacerbates the officials' difficulties.

10. Community recovery and the rebuilding of everyday life are difficult to pursue under the resource constraints of municipal administration, which limits the municipality's capacity to respond flexibly and effectively.

11. Structural and operational burdens manifested in the link between staffing problems and the deterioration in the health of municipal officials, as evidenced by prolonged overwork.

12. The administrative challenge of organising post-evacuation housing adds yet another layer of complexity, stretching local municipal capacities to their limits.

*Mr. Y*

1. In the context of inadequate national government oversight and the need for contractor internal controls, new and unfamiliar work emerges for municipal officials.

2. The absence of strong supervision forces municipalities to take on roles beyond their original scope, including the local government's responsibility for assessing and accounting for the appropriateness of projects.

3. The municipality is required to engage in human resource management, including tasks, to avoid hiring inappropriate people, which has become a significant local burden in the decontamination project.

4. New tasks in workforce management are tied to staff fatigue in the decontamination project due to insufficient human and financial resources.

5. The situation is further complicated by the Ministry of the Environment's lack of required expertise.

6. Personnel lacking the expertise to determine the direction of reconstruction and decontamination are assigned due to bureaucratic, compartmentalised decision-making.

7. Problems for laypeople, bureaucracy, and inadequate decision-making are apparent at the national level.

8. Reduced motivation due to incompetent national government officials among municipal officials inevitably occurs, and growing frustration with top-down decision-making becomes dominant.

*Mr. Z*

1. The uncertain future of the national decontamination plan and fluid national policy creates widespread confusion at the local level.

2. Confusion in municipal government due to the constant policy changes of the national government has become a persistent problem, while promises that are not kept because of changes in personnel have deepened frustration among municipal officials.

3. Municipal officials face difficulties mediating between the national government and the local population, often caught in the conflict between following the national reconstruction plan and prioritising local interests.

4. The municipal officials feel deepened distrust of the national side due to the national government's decontamination initiative being seen as just for show, rejection of ideas based on frontline feedback, hiding the real situation through staged site visits by politicians, and deception in government statements about thorough decontamination.

5. National government officials' views and attitudes that look down on municipal officials, combined with 'top management' limitations lacking knowledge of the local situation, create significant tensions between municipal and national government officials.

6. Distrust of bureaucrats' inability to understand the local situation spreads among municipal officials, accompanied by dislike of national government officials' views and attitudes and a lack of confidence in national government explanations.

7. The municipal officials consider the lack of conviction in the national government's views, attitudes, and explanations as a betrayal by the national government.

8. The national government shirks its responsibility by failing to keep records, forcing municipalities to bear the heavy burden of maintaining disaster records on its behalf, even as national government officials persist in corner-cutting on recordkeeping.

9. National government officials' statistical view of the disaster clashes with the role in communicating with residents in the intimate sphere that municipal officials are expected to perform.

10. Reduced trust in municipal government due to the collateral damage of distrust in the national government weakens the ties between residents and municipal officials.

11. The national government's dishonest attitude towards residents fuels rumours that municipal government officials give special treatment to their own neighbourhoods, leading to accusations and damaging public servants' pride.

12. Municipal officials are placed in a position unable to escape accountability to residents, torn between fairness as municipal government officials and attachment as residents.

13. Frustration with the snail's pace of decontamination and doubts about the never-ending process reflect the growing gap between policy and reality.

14. Municipal officials are concerned that the plan lacks a decontamination perspective to protect residents' lives upon their return.

15. Municipal officials regret that the most suitable site for waste disposal and temporary storage was not selected.

16. Compensation money used to suppress opinions, residents' money-grabbing behaviour, residents' greed, and residents' secretive attitude create an additional burden for municipal officials and deepen distrust between officials and residents.

17. For municipal government officials, the dysfunction of the "last bastion" of resident welfare undermined the very foundation of their professional pride.

18. Municipal officials feel a sense of emptiness in their work.

Note: The theories are presented for each interviewee in the order in which their narratives were provided. In the main text, however, the order has been rearranged for clarity.

work. I also had some knowledge of construction management and various paperwork. That may be a reason why I have been assigned to this department. However, when it came to the actual contracts with these companies, there were difficulties. In particular, there were cases where the Ministry of the Environment was the client, and in such cases the local authorities could not be involved in the details of the contracts."

(Narrative No.2 of Mr X in S1 Table).

Theories X1 and X5 suggested that MGOs may be dissatisfied with national radiation dose standards, as they are insufficient to alleviate residents' anxieties from the MGOs' viewpoint. Moreover, theory X9 indicated that such discrepancies between national and municipal ultimate objectives of the decontamination project can blur the direction of municipal work, and labour under such ambiguous purposes may lead to confusion among MGOs. The following narrative, included in theory X5, represents such objective discrepancies:

"There is a standard that evacuation orders will be lifted if the radiation level falls below 20 millisieverts per year. However, the government leaves the decision to lift the evacuation orders to the local authorities. In the end, it's just a complete handover. If the local government tries to persuade the residents and says, 'The decontamination is finished, the level has dropped to 20 millisieverts, so we're going to lift the evacuation order and you can go home,' do you think the residents will be convinced?"

(Narrative No.5 of Mr X in S1 Table).

Theories X6–8 and X10–12 described the expansion of municipal administrative duties and the excessive responsibilities imposed on MGOs. These theories suggest that MGOs must anticipate the expanded administrative roles awaiting them after the project's completion, such as lifting residents' evacuation orders, providing support to returnees, and securing housing, even while engaged in decontamination. The theories also suggested that although these roles should ideally be carried out in coordination with, or with assistance from, the national government, such collaboration and support may be far from adequate. The burdens of taking such roles are represented in the following excerpt for theory X12:

Interviewer: I think the role of your job will change, but what kind of...

Mr X: What I am most concerned about is that when the temporary accommodation is gone, there will be work to do to find out where the residents are going to go. During the evacuation, local government officials had to go through a lot of trouble to find temporary accommodation for the evacuees, such as rentals or inns. In the future, when the rent subsidy system for disaster victims disappears and they have to pay their own rent, and when there are no more temporary shelters, where will they go? Even if the local government were to provide housing for these people, such as reconstruction housing, the current financial and human resources of our local government would only allow the construction of a few dozen houses at most. In reality, the communities need thousands of houses, but there is no way we will be able to provide those.

(Narrative No.13 of Mr X in S1 Table).

## Mr Y

Eight theories derived from Mr Y's narrative illustrated the burdens that may be placed on the municipal level as a result of inadequate management by the national government and its officials.

Theories Y1–4 stated that, within the decontamination project, roles that should ideally be fulfilled by national ministries—such as the supervision and oversight of contractors/workers engaged in decontamination—are not always properly carried out and that these roles may consequently be transferred to MGOs. These theories indicate that these roles may lead to increased fatigue among MGOs because they are unfamiliar with these new responsibilities. The following

narrative illustrates the difficulty MGOs face when even tasks related to workforce quality control and public safety—such as verifying whether workers were inappropriate or unqualified—become part of the municipality's responsibilities:

> "The human resource management of the radiation decontamination workers by the Ministry of the Environment is not good. One of the difficulties in managing the decontamination work is that some dangerous individuals are found among the workers of decontamination companies. It does not matter whether they have a criminal record or not. Even if they have been in trouble in the past, people who have been socially sanctioned and are trying to get back on their feet should be employed. However, the human qualities of the staff have to be assessed. The Ministry of the Environment only does a cursory check."

> (Narrative No.1 of Mr Y in S1 Table).

> "The supervision by the ministry/national government does not always work properly for these contractors, the companies involved in the decontamination and reconstruction. We, the local government officials, have to manage it."

> (Narrative No.4 of Mr Y in S1 Table).

Theories Y5–8 suggested that difficulty in the decontamination project management for MGOs stems from the vertically segmented administrative system. The theories showed that national government officials with the expertise and knowledge required for the decontamination project are not assigned to municipalities, thereby hindering smooth progress and sound decision-making. The negative consequences of such management for MGOs were outlined: MGOs may become dissatisfied with the national government's top-down decision-making, which they perceive as ill-suited to on-site realities, leading to a decline in their motivation for their work. The following narrative in theory Y5 indicates the administrative system problems:

> "Well, the Ministry of the Environment is not an expert in civil engineering. It was not good that the Ministry of the Environment held the 'purse' (financial resources) in the name of environmental restoration because of the environmental pollution caused by the nuclear accident. Let's think about it. If farmland need to be decontaminated and restored, the Ministry of Agriculture, Forestry and Fisheries should take the lead, and if roads and buildings need restoration, it should be done by the Ministry of Land, Infrastructure, Transport and Tourism. Environmental restoration projects such as decontamination are essentially civil engineering work, and the Ministry of the Environment does not necessarily seem to have the required know-how."

> (Narrative No.5 of Mr Y in S1 Table).

### Mr Z

The eighteen theories generated from Mr Z's narrative highlighted a chain breakdown of trust—between the national government and residents, between the national government and MGOs, and between MGOs and residents—triggered by the perceived insincerity of the national government. In these theories, the focus shifted from institutional or managerial issues to the emergence of subjective distress.

Theories Z1–2, 4, 7, and 13 reflected anger toward the national government's superficial, insincere attitude toward reconstruction. MGOs, when confronted with the national government's policy inconsistency, unfulfilled promises, staged site visits, and inadequate information management, began to question whether the national government was truly engaging in reconstruction with genuine concern for the affected communities. The following narrative excerpt included in Theory Z4 exemplifies this sense of doubt held by MGOs toward the national government:

> "Sometimes, important government officials visit temporary storage facilities, but these facilities are built in 'clean' places. Ordinary temporary storage facilities are just heaps of stuff piled up. When important government officials visit,

the staff at these facilities show how well the site is maintained. They do this by showing the officials the sandbags to shield the area, the fences to block it off, the paving, the rainwater measures, the gas venting measures and the barricades. With such perfect maintenance, there is nothing to worry about. This maintenance should be standard for all temporary storage sites, but it's not. I don't want to say anything bad. Still, the facilities are very good, only in places where government officials/politicians can see them. That's not good. It's just making the media, including TV, look good. This doesn't show the true situation."

(Narrative No.2 of Mr Z in S1 Table).

As indicated in Theories Z3, 10–12, and 16, difficult relations with the national government were reported to have led to the deterioration of trust between residents and MGOs. These theories also describe a conflict in which MGOs must balance the interests of their local communities with their obligation to follow national policies on reconstruction and decontamination. MGOs thus find themselves caught between the national government and residents, struggling to mediate the relationship between them (Theory Z3). Furthermore, distrust and dissatisfaction among residents toward the national government often extended to municipal governments as collateral damage, weakening residents' trust in MGOs (Theory Z10). Theory Z16 suggested that as the bond between residents and MGOs weakens, residents may become reluctant to speak openly with municipal officials, thereby hindering administrative work. The following narrative excerpt from Theory Z10 illustrates the position of MGOs caught in such conflicts:

"No one in the community really trusts the effectiveness of the decontamination or the radiation levels in their food, even though the government says there is no problem. This is because they feel that the bureaucrats and politicians of the past have lied about everything, even before the nuclear accident. In this situation, the local people don't trust anything. Even what local administrators like us, who meet with local people on a regular basis, say is not trusted. It's as if the political distrust of the national government translates into distrust of the local government. Many people say that what other countries and inspectorates show about radiation is more trustworthy."

(Narrative No.12 of Mr Z in S1 Table).

Theories Z5–6 and Z8–9 addressed doubts about national officials and the bureaucratic system. Frustration arose from the tendency of national officials to make top-down decisions without sufficient understanding of local communities (Theory Z5); concerns emerged regarding the government's failure to preserve official records of the decontamination project (Theory Z8); and resentment grew toward national officials who treated the Fukushima situation as merely a statistical phenomenon rather than a lived reality (Theory Z9). The following narrative excerpt, which constitutes one of the components of Theory Z9, conveys this emotional reaction stemming from the disparity between the perspectives and sensibilities of national officials and those of MGOs:

"The national government's attitude is outrageous. They don't see the people. Even when something unusual happens in local areas, "Kasumigaseki"* carries on with administrative tasks as if nothing happened. Isn't the recent earthquake in Kumamoto a good example? What they're focusing on are just the figures and numbers on the paperwork."

(Narrative No.13 of Mr Z in S1 Table).

*Japanese bureaucracy or the administrative branch of the national government.

In Theories Z14–15 and Z17–18, the narratives expressed a sense of emptiness experienced by MGOs during their involvement in the decontamination project. This feeling appeared to stem from the realisation that their dedicated efforts were not necessarily effective in facilitating residents' return or in rebuilding their livelihoods (Theories Z14–15). As a

result, their professional pride—as those who had considered themselves the "last bastion" of resident welfare—was undermined (Theory Z17), leading ultimately to a profound sense of emptiness toward their work (Theory Z18). The narrative excerpt shown below, a component of Theory Z18, exemplifies this sense of emptiness:

> "When this kind of situation continues, it's exhausting. The national government can't be trusted, and the residents criticise us. It makes us worry about the future; what are we working so hard for? I feel empty."

(Narrative No.37 of Mr Z in S1 Table).

## Discussion

### National-municipal government relationship and MGOs' hardship: Individual-level findings

The first purpose of our study is to descriptively understand the nature and causes of subjective distress among MGOs.

The theory derived from Mr. X's experience captured (1) the background by which he came to engage in the newly introduced decontamination project despite limited preparedness, (2) the municipal confusion arising from the discrepancy between national standards and policies and the locally required support for residents and communities—namely, a misalignment in the project's underlying objectives, and (3) the excessive responsibilities placed on MGOs as municipal administrative functions expanded. Decontamination after radiological disaster is legally mandated to be managed by each municipality under the national government's plan [3]. However, the substantive control remains with the national government, requiring MGOs to implement decontamination while interpreting the national government's intentions within the constraints of limited autonomy. Previous studies have shown that limited discretion and ambiguous workloads can reduce motivation and worsen mental health among municipal employees [27–30]. It is therefore reasonable to assume that similar conditions imposed a significant burden on the officials in our study.

The theory informed by Mr Y's account revealed the possible burdens imposed on municipal governments by the national government's insufficient oversight. Notably, while the theory in Mr X's experience highlighted the organisational dysfunction embedded in government–municipality relations, the theory derived from Mr Y demonstrated how these organisational problems materialise in frontline operations of decontamination projects.

Building on this line of research, our findings suggest that managing decontamination workers may also be a significant psychological burden for MGOs. As reported in the media, some decontamination workers were linked to criminal groups, which raised concern among residents [30], potentially requiring MGOs to manage these workers to ensure public safety. Another study demonstrated a correlation between the management of poorly behaved workers and subjective distress for managers [31]. This responsibility likely increased mental health risks among MGOs, as demonstrated by a previous study showing a correlation between managing problematic workers and subjective distress among managers.

The theory in Mr Y's account also depicted the marginalisation of decision-making at the operational level, as well as MGOs' growing dissatisfaction and declining work motivation, resulting from the national government's failure to dispatch appropriate personnel. Japanese bureaucratic system is often described as highly segmented, with ministries operating within clearly defined jurisdictions [5]. This institutional structure tends to make cross-ministerial collaboration, if any, both difficult and less effective. Such systemic feature may result in the assignment of personnel who are inexperienced in collaborating with MGOs.

The theory derived from Mr. Z's provided more detailed explanations of ethical and emotional responses of MGOs working under the decontamination project than those of Mr. X and Mr. Y. MGOs perceived the attitudes of the national government and its officials toward reconstruction and decontamination as insincere, and they came to question the behavior and managerial practices of national officials who appeared to disregard the realities of frontline operations and the perspectives of municipal staff. At the same time, however, MGOs bore the consequences of the deteriorating

relationship between the national government and residents, which risked undermining their own trust-based relationships with the community. Ultimately, these experiences threatened the professional pride of MGOs—who had long regarded themselves as the "last bastion" of resident welfare—and could culminate in a profound sense of emptiness toward their work. We argue that this sense of emptiness, which arises in the final phase of MGOs' experiences, constitutes the essence of their subjective distress.

Past study indicated that exposure to workplace incivility, including disrespect, is associated with psychological stress [32]. In light of previous studies, it is reasonable to assume that the MGOs are at increased risk of mental health issues in coordinating with the national government during radioactive material decontamination projects. Following the FDNPP accident, residents' mistrust towards the national government primarily stemmed from the inadequacy and passivity in its risk and crisis communication policy [33]. A cycle of distrust appears to have been triggered by the government's initial inadequate response and subsequent failure to address the situation effectively. This led to damage in resident–MGO relationships, and the resulting mutual mistrust diminished MGOs' motivation to restore trust.

The threat to the belief that MGOs are the last line of defense for local residents' welfare is perceived as a blow to their pride as public servants. Pride arises from a positive self-evaluation when the organization to which one belongs receives attention and high praise [34]. Importantly, the erosion of such pride appears to lead to the consequence articulated by the officials themselves —namely, a sense of emptiness. Although the present study did not assess this "emptiness" using standard psychological scales, the fact that this term was explicitly mentioned in their narratives is a noteworthy finding. A previous study indicated that this feeling is associated with self-injury and suicidality [35]. In light of this, MGOs involved in decontamination projects may be at risk of mental health problems.

Through these individual-level analyses, it became evident that difficult relationships with the national government could ultimately lead to a profound sense of emptiness among MGOs. In line with the exploratory nature of this study, we integrate the theories derived from each interviewee and propose an exploratory conceptual model that offers a means of understanding these experiences as a process.

### Exploratory conceptual model of MGOs' subjective distress: Reorganising theories

The second purpose of the present study was to make an exploratory conceptual model of the process through which such subjective distress becomes manifest. Here, we present the exploratory conceptual model and explain it. Note that this conceptual model developed in this study does not aim to test predefined hypotheses but to illuminate key points at which subjective distress emerges in MGOs' work processes.

By reviewing and reorganising the theories derived from Mr X, Mr Y, and Mr Z, we hypothesised four stages in the time course that characterise the subjective distress experienced by MGOs when coordinating with the national government during the radioactive material decontamination project (Table 3). Stage 1, "the emergence of distrust and adversarial

Table 3. Conceptual stage model of subjective distress.

| Stage | Situation or problem | Explanation | Included theories |
|---|---|---|---|
| 1 | The emergence of distrust and adversarial orientations toward the national government | Doubts toward the national government arise from policy volatility, broken promises, staged site visits, and the concealment of information. | X1, 2–4, 5, 9 Y1-8 Z1-2, 4–9, 13 |
| 2 | The dilemma of occupying an intermediate position | A dilemma in which MGOs wish to remain sincere to both the national government and residents, yet find these expectations incompatible. | X6-8, 10–12 Z3 |
| 3 | The erosion of ties with local residents | A decline in morale and motivation caused by the unintended deterioration of relationships between MGOs and residents. | Z10-12, 16 |
| 4 | The threat to professional pride | An emotional outcome in which MGOs' sense of pride and mission becomes eroded, culminating in a feeling of emptiness. | Z14-15, 17–18 |

orientations toward the national government," refers to the initial doubts that MGOs developed about the government due to policy volatility, broken promises, staged site visits, and the concealment of information. Stage 2, "the dilemma of occupying an intermediate position," captures the tension MGOs felt as they sought to remain sincere and accountable to both the national government and residents, despite the incompatibility of these expectations. Stage 3, "the erosion of ties with residents," denotes the decline in morale and motivation brought about by the unintended deterioration of relationships between MGOs and the community. Stage 4, "the threat to professional pride," describes the emotional consequence in which MGOs' sense of pride and mission became increasingly undermined, culminating in a profound feeling of emptiness.

Stage 1 refers to the period when MGOs began to develop doubts and concerns regarding national-level officials and administrative systems through their direct involvement in coordination with the national government during the decontamination project. In the policy formulation process, it is crucial to incorporate diverse perspectives and local needs by establishing regular dialogue between central and local government officials, and by gathering input from municipal staff and residents through participatory methods such as surveys, workshops, and public forums [36]. Following the FDNPP accident, efforts based on these principles have already been implemented in Japan and reported to be effective [37]. A previous administration study reported that effective interagency collaboration can be promoted by clearly understanding a partner organisation's mission, values, and institutional norms, thereby increasing opportunities to align goals between agencies [38]. Strengthening coordination among ministries in the national government may also help cultivate appropriate experts for radioactive material decontamination and improve relationships between national and municipal governments, as well as those between national government officials and MGOs. Our model suggests that in the early phase of the post-radiation-disaster decontamination project, improvements in inter-ministerial communication, as well as communication between ministries and municipal governments, would likely play an important role in alleviating the mental burden on MGOs.

In Stage 2, the dilemma faced by MGOs becomes particularly evident, arising from their dual identity as both community residents and administrative staff, employees of the national or municipal government. Such dilemmas correspond to what prior research has conceptualised as role conflict—"the psychological tension arising from incompatible demands within one's role or between multiple roles" [39]—for which various mechanisms and countermeasures have been examined. Earlier studies have proposed several social and organisational strategies to mitigate role conflict, including improved scheduling and flexibility [40], clearer role definitions [41], and structural interventions such as revisions to organisational policies [42].

However, these social and organisational measures may not function effectively in the context of post-radiation-disaster decontamination projects and subsequent reconstruction efforts, given their inherent unpredictability and task complexity. In contexts where organisational improvement is limited or unavailable, coping at the individual level becomes especially important. Previous research has emphasised the necessity of providing individualised mental health support for MGOs facing uncertainty about their communities' future in the aftermath of a disaster [43].

Our model may be applicable for the support of the MGOs who were in a dilemma. While such individual-level mental health interventions are indispensable for supporting MGOs, it is important to recognise the risk that organisations may overly attribute the dilemma to "individual adaptability," thereby overlooking conflicts rooted in structural deficiencies or flawed policy design. In parallel with mental health support, additional measures aimed at preventing MGOs from falling into such dilemma states—such as refining communication techniques to enhance the quality of dialogue between national officials and residents, or deploying mediators other than MGOs—may contribute more fundamentally to preventing these problems.

In Stage 3, the loss of residents' trust in MGOs—and the resulting decline in morale and motivation—is described. Even when MGOs strive to fulfil their responsibilities while navigating the role conflict outlined in Stage 2, such efforts do not necessarily lead to positive outcomes; instead, they may result in a weakening of the ties between MGOs and residents. Our model suggests the need for an approach to improve morale and motivation among MGOs, which are influenced by exacerbated relationships with residents. From the perspective of public service motivation theory [29], which emphasises

that public trust is essential for sustaining morale and resilience among public servants, this erosion of trust represents an undesirable consequence for MGOs. Appleby-Arnold et al. [44] presented evidence that the implementation of disaster-specific mobile applications featuring functions that allow citizens to provide information—such as disaster vulnerability of an area, alert, and emergency situation—to authorities can foster a sense of shared responsibility between municipal governments and residents. This, in turn, helps overcome citizens' perceptions that they are distrusted by authorities and contributes to rebuilding trust between them. Leveraging new media and information and communication technologies to promote bidirectional communication—not only from municipalities to citizens, but also from citizens to municipalities—may serve as an effective measure for alleviating the distress experienced by MGOs in Stage 3. Importantly, our findings demonstrate that MGOs may be subject to negative evaluations from residents despite their persistent efforts to disclose information regarding radioactive material decontamination projects. Therefore, rather than placing further communicative burdens on MGOs, the improvement strategy proposed here is intended as a realistic and contextually grounded option for supporting trust-building between municipality and residents, without adding new demands on MGOs or positioning itself as a substitute for existing national policies.

Stage 4 represents the final phase, in which MGOs experience a sense of 'emptiness' as the core manifestation of their subjective distress. The implication of our model, which suggests that a sense of emptiness may be positioned at the final stage of the process of subjective distress among MGOs, is the importance of equipping mental health support with a diverse range of approaches to address this sense of emptiness. Because clinical outcomes of emptiness include suicide [35], the importance of addressing such emptiness is evident. However, there remains a lack of well-established psychological intervention strategies specifically targeting emptiness. On the other hand, previous studies on emotional labour may offer useful insights: enhancing emotional self-management skills [45], ensuring access to personal resources that enhance work engagement, such as transformational leadership [45], and fostering trust-based relationships with colleagues in the workplace [46] are all considered promising strategies to reduce or prevent the sense of meaninglessness that underlies a feeling of emptiness associated with one's work. Even in the absence of psychiatric or psychological treatment options specifically tailored to the subjective distress stemming from the feeling of emptiness among MGOs during decontamination projects, these occupational health management approaches may provide effective means of prevention and/or alleviation.

Although this exploratory conceptual model is derived from a small sample of three MGOs in Japan, the fact that the duties of municipal government officials are legally prescribed [1,2] suggests that difficulties in national–municipal relations similar to those described in this study could reasonably arise in future radiation-related disasters. Importantly, because the roles and responsibilities of municipal (local) officials are generally defined by national law not only in Japan—the target country of this study—but also in many other countries/societies, it is reasonable to argue that the insights presented here may be transferable to countries and societies outside Japan which may face a radiation disaster in the future.

### Limitations

The present study has four limitations. First, it employed a cross-sectional design and merged interview data collected over three years. Consequently, it was not possible to analyse the dynamic changes in subjective distress and support needs over time. Future studies should adopt a longitudinal design and conduct a more in-depth analysis of MGOs' experiences during the reconstruction following a large-scale nuclear disaster.

Second, this study did not fully incorporate perspectives from organisational science in our discussion. While psychological or occupational health support may help alleviate MGOs' subjective distress, more fundamental solutions require addressing systemic issues, such as Japan's administrative and labour systems, as well as communication strategies between national and municipal governments. These remain important areas for future research to explore. It should be particularly emphasised that strengthening both macro-level measures—grounded in institutional design and organisational science—and micro-level, interpersonal interventions—such as occupational mental health and psychological support—will enhance societal resilience to future disasters and safeguard MGOs as workers.

Third, the interviewer had been cultivating a relationship with the interviewees as part of a health support project, which may have introduced potential bias into the narrative content. The interviewees may have exaggerated their challenging experiences in an attempt to elicit assistance, or conversely, may have presented a modest narrative to indicate that they had no health problems. Future studies need to conduct surveys targeting MGOs in a variety of roles in order to gain valuable insights into the mental health issues within this demographic in the aftermath of a radiation disaster.

Fourth, it should be acknowledged that the difficulties disclosed by the interviewees—such as their subjective distress or their strained relationships with the national government—represent only part of MGOs' experiences and may not capture them in their entirety. It is possible that interviewees found it easier to speak about their difficult relationships with the government; if so, we must consider the possibility that some information remains undisclosed. MGOs' experiences are embedded in a broader set of social relationships—including those with residents, colleagues, and other relevant actors—which were not fully examined in the present analysis. There is an appropriate time to elicit and understand MGOs' experiences. In this study, interviewees were able to share their stories because a long period had passed since the disaster, and conditions were in place to ensure that no individuals could be identified. However, building a longer-term relationship of trust between researchers and participants may create further opportunities for a richer and more in-depth qualitative inquiry.

## Conclusion

The purpose of this study is to descriptively understand the nature and causes of subjective distress among MGOs and to make an exploratory conceptual model of the process through which such subjective distress becomes manifest. The details of the subjective distress experienced by MGOs, as identified through the qualitative method SCAT, were summarised for each interviewee as follows: the structure, in which inconsistencies in national reconstruction and decontamination designs impose burdens at the operational level, leads to the exhaustion of individual MGOs (Mr X); the burdens arise on the municipal level as a result of the inadequate management of national government and officials (Mr Y); and a chain breakdown of trust associates the emergence of subjective distress—between the national government and residents, between the national government and municipalities, and between municipalities and residents (Mr Z). We further modelled the process through which subjective distress emerges among MGOs by organising these narrative-based theories along a temporal trajectory. In this exploratory conceptual model, the progression toward subjective distress is explained by four stages: the emergence of distrust and adversarial orientations toward the national government; the dilemma of occupying an intermediate position; the erosion of ties with residents; and the threat to professional pride.

Although this model was developed based on the recovery process after the FDNPP accident, and is therefore contextually grounded in the dynamics of national–municipal relations in Japan, it may be transferable to other cases, given that decontamination work is an essential process following nuclear disasters. Once a nuclear disaster occurs and affects a municipality, extensive coordination between national and municipal governments is typically required to undertake tasks such as confirming residents' intentions to return, providing livelihood support, and facilitating housing assistance. It is reasonable to assume that MGOs may experience subjective distress during such coordination. The utility of this model will be maximised when institutional arrangements and specific individual-level support strategies are tailored to each stage, taking into account the available resources in the specific region and time in which the model is applied. Consequently, we are convinced that strengthening both macro-level measures—grounded in institutional design and organisational science—and micro-level interventions—such as occupational mental health and psychological support—is essential for reducing subjective distress among MGOs and advancing post-disaster reconstruction efforts.

## Supporting information

**S1 Table. SCAT analysis form.** This analysis form is an official tool designed to progressively abstract qualitative data (text) and construct theory through the following four stages: (1) noteworthy words or phrases from the text, (2) paraphrases of (1), (3) concepts from outside the text that account for (2), and (4) themes and constructs developed in consideration of context. (XLS)

## Acknowledgments

We would like to express our sincere gratitude to the interviewees for their cooperation with this study.

## Author contributions

**Conceptualization:** Tomoo Hidaka.

**Data curation:** Tomoo Hidaka.

**Formal analysis:** Tomoo Hidaka.

**Funding acquisition:** Tomoo Hidaka.

**Investigation:** Tomoo Hidaka.

**Methodology:** Tomoo Hidaka.

**Project administration:** Takeyasu Kakamu.

**Resources:** Takeyasu Kakamu.

**Supervision:** Takeyasu Kakamu.

**Validation:** Hideaki Kasuga, Yuko Suzuki, Toshihiro Terui, Shizuka Kawamoto, Tatsuya Sato.

**Writing – original draft:** Tomoo Hidaka.

**Writing – review & editing:** Tomoo Hidaka, Hideaki Kasuga, Yuko Suzuki, Toshihiro Terui, Shizuka Kawamoto, Tatsuya Sato.

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
