## [Decision Letter · Decision Letter 0]

7 Oct 2025

Dear Dr. Hidaka,

Thank you for submitting your manuscript to PLOS ONE. After careful consideration, we feel that it has merit but does not fully meet PLOS ONE’s publication criteria as it currently stands. Therefore, we invite you to submit a revised version of the manuscript that addresses the points raised during the review process.

We look forward to receiving your revised manuscript.

Kind regards,

Sakae Kinase, Ph.D.

Academic Editor

PLOS ONE

Journal Requirements:

2. In the online submission form, you indicated that [The data are not publicly available due to their containing information that could compromise the privacy of research participant. The data that support the findings of this study are available on request from the corresponding author, Tomoo Hidaka.].

3. Please include captions for your Supporting Information files at the end of your manuscript, and update any in-text citations to match accordingly. Please see our Supporting Information guidelines for more information: http://journals.plos.org/plosone/s/supporting-information....

4. We note that this data set consists of interview transcripts. Can you please confirm that all participants gave consent for interview transcript to be published?

If they DID provide consent for these transcripts to be published, please also confirm that the transcripts do not contain any potentially identifying information (or let us know if the participants consented to having their personal details published and made publicly available). We consider the following details to be identifying information:

- Names, nicknames, and initials

- Age more specific than round numbers

- GPS coordinates, physical addresses, IP addresses, email addresses

- Information in small sample sizes (e.g. 40 students from X class in X year at X university)

- Specific dates (e.g. visit dates, interview dates)

- ID numbers

Or, if the participants DID NOT provide consent for these transcripts to be published:

- Provide a de-identified version of the data or excerpts of interview responses

- Provide information regarding how these transcripts can be accessed by researchers who meet the criteria for access to confidential data, including:

a) the grounds for restriction

b) the name of the ethics committee, Institutional Review Board, or third-party organization that is imposing sharing restrictions on the data

c) a non-author, institutional point of contact that is able to field data access queries, in the interest of maintaining long-term data accessibility.

d) Any relevant data set names, URLs, DOIs, etc. that an independent researcher would need in order to request your minimal data set.

For further information on sharing data that contains sensitive participant information, please see: https://journals.plos.org/plosone/s/data-availability#loc-human-research-participant-data-and-other-sensitive-data

If there are ethical, legal, or third-party restrictions upon your dataset, you must provide all of the following details (https://journals.plos.org/plosone/s/data-availability#loc-acceptable-data-access-restrictions):

1. A complete description of the dataset

2. The nature of the restrictions upon the data (ethical, legal, or owned by a third party) and the reasoning behind them

3. The full name of the body imposing the restrictions upon your dataset (ethics committee, institution, data access committee, etc)

4. If the data are owned by a third party, confirmation of whether the authors received any special privileges in accessing the data that other researchers would not have

5. Direct, non-author contact information (preferably email) for the body imposing the restrictions upon the data, to which data access requests can be sent.

Additional Editor Comments:

Your paper has been carefully considered by two referees. The referees' comments indicate that some fundamental revisions are necessary before the paper can again be considered for publication in PLOS One. Please carefully consider the comments, it is recommended that you revise the paper accordingly.

Reviewers' comments:

Reviewer's Responses to Questions

**Comments to the Author**

1. Is the manuscript technically sound, and do the data support the conclusions?

Reviewer #1: Partly

Reviewer #2: No

2. Has the statistical analysis been performed appropriately and rigorously?

Reviewer #1: N/A

Reviewer #2: N/A

3. Have the authors made all data underlying the findings in their manuscript fully available?

Reviewer #1: Yes

Reviewer #2: Yes

4. Is the manuscript presented in an intelligible fashion and written in standard English?

Reviewer #1: Yes

Reviewer #2: Yes

Reviewer #1: The authors investigated the impact of decontamination efforts on municipal officials and examined the structure of their subjective distress. This study is original and meaningful, but there are significant methodological issues that need to be addressed.

The authors should consider the following points:

1. There is no description of the analysis procedure, including what SCAT is. The manuscript does not explain how the authors analyzed the data and derived their results. The Methods section should specify what the segments (theoretical segments), components, and themes in the results represent, and how they were generated. In addition, is the term "theoretical segment" appropriate? In SCAT, a “segment” refers to a piece of data entered into a single row of the analysis spreadsheet (Otani T. Paradigm and Design of Qualitative Study: From Research Methodology to SCAT. The University of Nagoya Press, 2019, p. 280).

2. What are the four sentences (theoretical segments) in Table 2? They resemble “storylines” in SCAT. However, the purpose of the analysis is to generate theory, and the results should be theoretical rather than storylines (Otani T. Paradigm and Design of Qualitative Study: From Research Methodology to SCAT. The University of Nagoya Press, 2019, p. 311).

3. Another concern is that a single theoretical segment (which appears to be a storyline) contains themes derived from multiple participants. In SCAT, each interview preserves its own contextual specificity, and therefore three interviews should yield three individual analytic results. While it is acceptable to later discuss the findings from four perspectives, the analysis itself should first be conducted for each participant individually, and only at the discussion stage should the results be reorganized into four perspectives (Otani T. Paradigm and Design of Qualitative Study: From Research Methodology to SCAT. The University of Nagoya Press, 2019, p. 367).

4. The Introduction should include at least a minimal explanation of the decontamination activities conducted in Fukushima.

Reviewer #2: 1. General comments

This is a valuable and study investigating the psychological burden experienced by municipal governmental officials who play a crucial role in recovery from nuclear disasters placed importance on human emotions. Testimonies from these officials can offer valuable insights not only for preparing for future disasters but also for addressing current ongoing challenges. However, this is only useful when it provides substantive findings that ensure the quality of research methods and analysis.

This study presents five perspectives, focusing particularly on the complexity of the relationship between municipal governmental officials and the national government not mentioning contributions from Fukushima Prefectural government officials. However, these “relationship challenges” tend to be easily expressed in society. Consequently, it may simply be that easily articulated grievances are surfacing. Furthermore, the paper does not present the underlying structural social context and administrative science perspective for the specific challenges discussed. Presenting analysis results without deeper exploration risks merely deepening divisions and hindering constructive progress. While resource constraints are emphasized, decontamination costs amount to approximately 4 trillion yen, representing a significant scale of response costs for an accident-causing environmental pollution compared to other cases. However, such comments may be perceived by the authors—who are psychologically attached to the research subjects—as an unethical attitude that fails to understand the suffering of the affected municipalities, as evidenced by the recorded testimonies. Therefore, it would be better to logically present the structure that inevitably leads local government employees to harbor dissatisfaction with the national government. Furthermore, while individualized responses tailored to personal psychological states may be effective in specific situations, this issue is fundamentally societal in nature, raising concerns that this perspective may be being overlooked.

Addressing the initial missteps in the nuclear accident response is crucial, yet the dissatisfaction with Fukushima prefectural government and Fukushima medical university is entirely unmentioned. This raises questions about the survey's bias (selecting issues that are easier to discuss with the respondent; a cynical view might interpret this as the authors using the respondent to voice their own dissatisfaction with the national government). Therefore, a careful, fact-based approach is essential, and the questions raised by municipal government officials must also be addressed.

Post-nuclear disaster responses present challenges. Regarding seafood exports, establishing an external common enemy enhanced support for fisher-persons in Fukushima (though it was fisher-persons outside Fukushima prefecture who suffered damage due to lost export opportunities). However, achieving fair risk distribution domestically is likely to be considerably difficult. A paper grounded in stronger principles and factual evidence could contribute to fostering solidarity across nations.

The authors emphasize falsifiability, so the reviewer attempted to comment on the interpretation of the evidence. However, the paper belongs to the authors, and it is best to disregard unfair and unproductive comments stemming from the reviewers' lack of understanding.

I welcome sharing my comments with those who assisted with the survey and other staff members. I recommend obtaining approval from each municipality at least for the final draft to be submitted.

2. Title

A) If authors believe there are limitations to generalization, why not explicitly state that it is a case study?

B) It states that “radiation decontamination,” but what was decontaminated was not radiation but radioactive material; the term “radiation decontamination” is not rationale.

3. Abstract

A) Line 35: “may be misaligned with local needs;”

This is an ambiguous expression. As it lies at the core of the issue raised in this paper, it is essential to present the factual basis within the main text. Otherwise, it cannot be considered a scientific paper.

B) Line 39: “limited human and financial resources”

The Japanese government's decontamination costs amount to over 12 billion Euro with local governments outside Fukushima also providing personnel support (which has had various impacts on dispatched staff). While it may not be appropriate to directly compare each situation, discussions must take into account how local governments have responded to other environmental contamination incidents both domestically and internationally.

4. Introduction

A) Line 49: “play key roles”

Citizens are also key contributors to decontamination efforts in their communities, particularly when establishing temporary storage sites. In Date City, which led decontamination activities early on, progress was made through collaboration between citizens and municipal governmental officials. This initiative is outstanding and offers tremendous value for learning. However, city employees who provided favors to researchers have faced disciplinary action. Citizens who denounced this outcome remain entirely unconvinced. Understanding these complex circumstances should be a fundamental prerequisite.

B) Line 60: “mandated by the national government”

Is this an appropriate relationship between the national government and municipal governments? Hasn't it become necessary to respond to residents' needs?

C) Line 65: “little is known”

It is advisable to derive research hypotheses based on prior studies and statistical data, incorporating existing knowledge. The authors also have access to information referencing document number 41. In addition to the study listed below, surveys were also conducted by the Fukushima Prefectural and Municipal Workers Union.

https://doi.org/10.11361/reportscpij.21.4_439

D) Line 77: “These characteristics may be the reasons for the reluctance of the affected MGOs to disclose their views regarding the national government”

It is thought that a considerable number of local government officials are involved in this work. Even with anonymity guaranteed, was it considered difficult to secure their cooperation for the survey? I believe there have been a certain number of instances, at least within Japan, where local government employees have presented their views to public.

E) Line 79: “their anonymity must be protected;”

Respect for the autonomy of the parties involved is necessary.

F) Line 85: “in a difficult relationship with the national government.”

What are the administrative science-based reasons for the difficulty?

G) Line 88: “potential countermeasures”

Given the amount of time that has passed, authors should be able to make proposals based on scientific evidence that incorporates all previous attempts.

H) Line 89: “hypothetical model”

Readers are interested in the theoretical basis for setting hypotheses and how those hypotheses were verified.

5. Methods

A) Line 104

Is the presentation of gender and attributes in accordance with the subject's wishes?

B) Line 243: “Line 104”

It is self-evident that even a very limited interview with a single subject can yield meaningful insights. Furthermore, how can the insufficient recruitment of subjects be explained in light of this study's objectives? Incidentally, is the main analysis in this paper supported by the subjects of this research?

6. Results

A) Line 260: “illustrated the divergence in priorities between the national government and local stakeholders.”

What differences were there in terms of priority? And how did those differences affect the “ultimate decontamination goals”?

B) Line 271:

While coexisting with the temporary storage area involved various ingenious solutions, what function did the interviewer's responses to rhetorical questions aim to serve, considering these efforts?

Can anyone other than Japanese people even grasp the significance of this rhetorical question in the first place?

Does this indicate the authors' interpretation that it has become a convenient excuse to avoid confronting the underlying difficult issues?

C) Line 272:

Are authors assuming that readers would think, “It goes without saying that just because the scheduled period has ended doesn't mean the work is finished”?

D) Line 290:

Considering the context from the previous sentence, do you think this passage is pointing out that they don't understand the know-how regarding what?

E) Line 293:

Is this pointing out a lack of opportunities to communicate with suitable personnel, or is it highlighting a different issue? If it's the latter, what kind of improvement ideas might be considered? What elements do the authors speculate this uncertainty pertains to? Regarding the transition in the final sentence here, is the reader expected to recognize that the logic doesn't hold?

F) Line 310:

Additional information regarding support staff from other municipalities would also be helpful.

G) Line 323:

At that time, this was a concern frequently voiced by residents, but it would also be helpful to add how businesses and other establishments responded to it.

H) Line 340:

If the information provided by the Ministry of the Environment and local governments was insufficient to counter the spread of misinformation, it would be better to explain why that was the case.

https://www.env.go.jp/en/

7. Discussions

A) Line 541:

It would be better to provide specific examples of instances where government agencies provided information that was not “a realistic and contextually grounded alternative.”

8. Conclusions

A) Line 19, Page 16: “The of ages of artifacts…”

The first “of” would be unnecessary.

B) In the conclusion, authors should concisely state what this study has revealed. From this perspective, the introduction must state what research hypothesis this study aims to clarify and why verifying that hypothesis is important.

8. Data availability statement

While it is considered to be anonymized, it would be beneficial to express opinions on the disclosure of speech records from the perspective of open science.

9. Table 2

A) “national radiation exposure standards”

Considering how it aligns with IAEA GSR Part 3, it would be better to explain why municipal government officials felt dissatisfied.

B) “through staged site visits by politicians”

It would be better to explain what intricate schemes were involved, taking into account the disclosure of facts regarding the bureaucrats' briefing to politicians. In particular, it would be better to clearly show how the mayors and the prefectural governor countered such treatment.

C) It would be good to highlight what protests Fukushima Prefectural Assembly members and members of local assemblies within Fukushima Prefecture carried out, and how Japanese citizens outside Fukushima Prefecture ignored those protests.

D) “fluid national policy”

It would be helpful to include supplementary information on the mechanisms by which “fluid national policy” was formed.

It would be good to also highlight the challenges of flexibly revising plans as the situation evolves.

E) “The plan lacks a perspective on decontamination for the sake of residents' lives when they returned”

The explanation of how this situation came about is also important. In this explanation, the lawsuit filed by Tsushima in Namie Town, Futaba District, can also be cited as an example (the plaintiffs' group's website—which was inaccessible for a long time, yet the plaintiffs endured this—contains a significant error in the English translation of the plaintiffs' group's name, which remains uncorrected).

F) “frustration at the snail's pace of decontamination”

If he was explaining a delay in the plan, it is helpful to supplement authors’ explanation with why the delay occurred and how much time was lost.

G) “decontamination by the national government's initiative as just for show.”

It would be helpful to explain the power balance structure that prevented the municipal government from taking the lead.

H) “the never-ending process”

It would be helpful to explain what process is being referred to and why it is endless.

I) “The dishonest attitude of the national government towards residents”

It would be helpful to have specific examples.

J) “the lack of conviction in national government explanations”

It would be good to indicate what kind of “conviction” is expected.

K) “These risks exacerbate by the limitations of 'top management' without the knowledge of the local situation, leading to wrong decisions.”

It would be better to clarify what ‘top management’ refers to.

L) “looking down”

It would be better to clarify what ‘looking down’ refers to.

M) “a lack of respect for our expertise and efforts”

It would be better to clarify what 'expertise and efforts' refers to.

N) “not keeping records”

It would be better to clarify what ' not keeping records' refers to.

O) “keep disaster records”

It would be helpful to explain why the difficulty of this administrative work cannot be overcome through the use of ICT.

P) “by corner-cutting by national government officials on record”

Here, a municipal official was criticizing national government officials for failing to keep records, was not it? By automatically recording the national government officials' failure to keep records, could this effectively take over the task of record-keeping from the national government officials?

Q) “promises that are not kept because of changes in personnel”

It would be beneficial to also present the Japanese government's response to the written question submitted by the Fukushima-elected Diet member, which demanded an explanation for why the promises were not kept, after presenting the official documents detailing what was agreed upon between national government officials and municipal government officials.

R) “Their approach, including the use of compensation money used to suppress opinions, does not address the root issues.”

Compensation money is not intended for decontamination efforts. It would be helpful to explain what ‘suppress’ refers to.

S) “the ideas from frontline feedback”

It would be better to clarify what ' the ideas from frontline feedback ' refers to.

T) “the repeated betrayal by the national government”

It would also be beneficial to show how government agencies conducted briefings for Diet members elected from Fukushima Prefecture, as this has already been attempted by Fukushima residents.

U) “limited autonomy.”

If the inherent “autonomy” is impaired, it would be helpful to have an explanation as to why.

V) “excessive responsibility to plan and implement the timing of the lifting of the evacuation”

It would be helpful to have an explanation of what constitutes ‘excessive’ responsibility for matters that should fall under the jurisdiction of municipal government officials.

W) “conflicting interests and priorities”

It would be better to explain who is in conflict with whom. FMU experts are also contributing to resolving the conflict taking into account the positive precedents set overseas, aren't they?

X) “The reduced trust damages the relationship between municipal government and residents, and makes secretive attitude of local residents leaders”

The attitude of local residents' leaders is a crucial factor in addressing the issue, and it would be helpful to clarify what “secretive” refers to here. Additionally, an explanation of “local residents' leaders” would be beneficial.

Y) “the most suitable site for waste disposal and temporary storage is not selected.”

Given that this location is optimal and does not compromise fairness, it would be beneficial to supplement this by explaining what issues existed in the decision-making process.

Z) “Compounding these challenges is the money-grabbing behaviour by residents.”

It would be helpful to have supplementary information on the relationship between this assignment and Stage 1.

.

Reviewer #1: No

Reviewer #2: No

---

## [Author Response · Author response to Decision Letter 1]

26 Nov 2025

We would like to express our sincere gratitude for the careful and insightful reviews. The reviewers’ expert comments have been extremely helpful in improving the manuscript.

Before providing our point-to-point responses to each comment, we would like to outline three major revisions made in the revised manuscript:

1) In accordance with the editor’s instructions, we revised the manuscript to ensure compliance with the appropriate formatting requirements.

2) Following Reviewer #1’s concerns regarding the SCAT analytic procedures, we fundamentally reconsidered the analysis; as a result, the previously presented five-stage model has been reorganised into four stages.

3) In response to Reviewer #2’s observation that the term “radiation decontamination” was unnatural, we revised the title accordingly.

All revisions in the manuscript are highlighted in yellow. As these adjustments entailed substantial changes to the paper’s overall content—especially the Results section, which has been entirely rewritten, along with the corresponding updates to the Discussion—we recognise that this imposes an additional burden on the reviewers during re-evaluation. We kindly ask for your understanding that these extensive revisions were necessary to address the reviewers’ comments adequately.

[Editor]

Q1. Please ensure that your manuscript meets PLOS ONE's style requirements, including those for file naming. The PLOS ONE style templates can be found at…

A1. Thank you for your comment. We have read the guidelines and revised our manuscript. If any further aspects require revision, we would appreciate your guidance.

Q2. In the online submission form...All PLOS journals now require all data underlying the findings described in their manuscript to be freely available to other researchers, either 1. In a public repository, 2. Within the manuscript itself, or 3. Uploaded as supplementary information...

A2. The interview scripts were included as a supplementary file with all personally identifiable information removed. Therefore, the present case corresponds to option 3 “Uploaded as supplementary information”.

Q3. Please include captions for your Supporting Information files at the end of your manuscript, and update any in-text citations to match accordingly. Please see our Supporting Information guidelines for more information: http://journals.plos.org/plosone/s/supporting-information.

A3. We have added the caption to the supporting information (in manuscript p.37)

Q4. We note that this data set consists of interview transcripts. Can you please confirm that all participants gave consent for interview transcript to be published?...

A4. The interviewees provided consent for the interview transcripts to be published. However, the previous manuscript included specific dates that could potentially lead to personal identification. To ensure anonymisation, we have revised the description of the interview schedule in the Methods section as follows: ‘We conducted one interview with Mr X and Mr Y in 2015, and five interviews with Mr Z between 2016 and 2017.’ (p. 7)

[Reviewer#1]

Q1. There is no description of the analysis procedure, including what SCAT is. The manuscript does not explain how the authors analyzed the data and derived their results. The Methods section should specify what the segments (theoretical segments), components, and themes in the results represent, and how they were generated. In addition, is the term "theoretical segment" appropriate? In SCAT, a “segment” refers to a piece of data entered into a single row of the analysis spreadsheet (Otani T. Paradigm and Design of Qualitative Study: From Research Methodology to SCAT. The University of Nagoya Press, 2019, p. 280).

A1. We appreciate the reviewer’s comment. The term ‘segment’ was indeed a mistranslation and therefore inappropriate; we have removed it and replaced it with more suitable expressions throughout the manuscript—for example, ‘narrative excerpts’ instead of ‘speech segments’, and ‘theory from Mr X/Y/Z’ instead of ‘theoretical segment’.

Regarding SCAT, we acknowledge that the explanation in the previous version, although present in the Qualitative Analysis subsection of the Methods section, was insufficient. We have therefore added the following information to clarify our analytical procedure (pp. 8-9):

“The verbatim transcripts of all the interviews were analysed cross-sectionally using Steps for Coding and Theorisation (SCAT), a qualitative data analysis method for inductively summarizing narratives, as described in previous studies [19-21]. SCAT has the advantage of extracting important excerpts from the original narrative and gradually abstracting them, ultimately deriving concepts while considering the meanings embedded in the original narrative. Specifically, we used the following seven steps: (1) extracting narratives that were considered important and rendering them into text; (2) selecting noteworthy words or phrases from the text (Step 1) to set the focus of the analysis; (3) paraphrasing the words and phrases extracted in Step 2 to convert the interviewee’s unique expressions into analyzable and interpretable forms; (4) identifying underlying regularities such as cause-and-effect relationships, sequential patterns, and/or tendencies within the data by using concepts from academic or professional domains that account for paraphrased texts in Step 3; (5) describing the core meanings expressed in the original narratives in abstract terms as concise headings—composed of nouns or noun phrases and referred to as “themes”—based on the results of Steps 2–4, in a way that generates novel conceptual constructs; (6) constructing a storyline that explains the interviewee’s experiences by using all themes generated in Step 5, supplementing them with additional wording where necessary; (7) by articulating the storyline in a generalized and unified descriptive form, describing the "theory" which is the brief and declarative expression what can be claimed based on the analysis of the text. The theory generated in Step 7 is the outcome of SCAT. Note that SCAT analysis is, in principle, conducted for each interviewee; accordingly, in this study as well, three distinct theories, corresponding to the number of interviewees, were ultimately generated.”

In accordance with SCAT’s formal procedures, we have also revised our analytical steps. Therefore, as noted above, please be aware that a separate theory was generated for each interviewee.

Q2. What are the four sentences (theoretical segments) in Table 2? They resemble “storylines” in SCAT. However, the purpose of the analysis is to generate theory, and the results should be theoretical rather than storylines (Otani T. Paradigm and Design of Qualitative Study: From Research Methodology to SCAT. The University of Nagoya Press, 2019, p. 311).

A2. Thank you for your comment. The analytic procedures were inappropriate. As the reviewer correctly points out, in SCAT, ‘theory’ refers to a written account developed from the storyline, characterised by generality, predictability, and coherence (source: p. 324 of Otani’s work cited by the reviewer). As described below, we have re-analysed the data and, accordingly, both the content and the presentation of the resulting theories have been substantially revised.

The number of theories ultimately generated is now reported in the Qualitative Analysis subsection of the Methods section (p.9):

“...as a result, 38 descriptions of theories were identified: 12 from Mr X, 8 from Mr Y, and 18 from Mr Z. These theories and underlying themes and storylines...”

In addition, the theories—conceptually distinct from the storyline—are explicitly presented in Table 2 as the results of our analysis. These are reported separately for each interviewee (Table 2, pp. 39–42). With these revisions, the content of the previous Table 2 has been replaced with an updated version.

Q3. Another concern is that a single theoretical segment (which appears to be a storyline) contains themes derived from multiple participants. In SCAT, each interview preserves its own contextual specificity, and therefore three interviews should yield three individual analytic results. While it is acceptable to later discuss the findings from four perspectives, the analysis itself should first be conducted for each participant individually, and only at the discussion stage should the results be reorganized into four perspectives (Otani T. Paradigm and Design of Qualitative Study: From Research Methodology to SCAT. The University of Nagoya Press, 2019, p. 367).

A3. The reviewer’s concern is valid. In accordance with SCAT’s formal procedures, we have re-conducted our analysis. We first generated a theory for each interviewee and reported these in the Results section. We then reorganised the theories derived from each interviewee to construct a model inferred from these analytic outcomes. Revisions have been made to document these analytical and interpretive procedures clearly.

First, we have added the following paragraph at the end of the Qualitative Analysis subsection in the Methods section, describing the creation of the model through reorganisation of the results (lines 210-217, p. 10):

“In the original SCAT procedure, the analysis is conducted and reported for each interviewee. However, because this study aimed to identify overarching patterns across participants, the researcher conducted an additional interpretive integration process that is not part of the standard SCAT procedure. Specifically, the individual theories generated through SCAT were further examined, compared, and conceptually grouped into higher-order categories through interpretive abstraction. This supplementary step was conducted in the discussion section to enhance integrative understanding while maintaining fidelity to the original SCAT principles by developing a hypothetical model of the process by which MGOs’ subjective distress becomes manifest.”

Second, because a separate theory was generated for each interviewee, we have added a new subsection to the Discussion section titled ‘National–municipal government relationship and MGOs’ hardship: individual-level findings’ (p. 20), in order to discuss the findings derived from each interviewee’s theory. We have also provided detailed explanations in this subsection (lines 448–456, 464–468, 487–498, and 520–523).

Third, we have provided a detailed explanation of the model obtained through the reorganisation of the theories in the subsection ‘Hypothetical model of MGOs’ subjective distress: Reorganising theories’ (p. 23). These represent the study’s integrated findings, and their four-step structure is summarised in Table 3. In the main text, new explanations have been added in lines 531–543, 545–547, 558–560, 562–569, 571–573, 578–584, 586–592, 606–607, and 620–627.

Please note that we have clearly distinguished between the theory generated by the standard SCAT procedure and the model derived from our interpretive (conceptual) analysis.

4. The Introduction should include at least a minimal explanation of the decontamination activities conducted in Fukushima.

A4. Thank you for your comment. We have added the explanation to the first paragraph of the Introduction section (lines 50-54, p.3):

“The decontamination project was a large-scale program aimed at removing radioactive materials released into the environment following the FDNPP accident, and it was an essential process for enabling residents to return and for promoting regional reconstruction[3]. Under the government's leadership, municipalities affected by contamination were legally obligated to engage in decontamination work[3]. Therefore...”

[Reviewer#2]

Before reporting the specific revisions made to the manuscript, please allow us to offer an overall, dialogic response to your comments.

Comment 1:

This study presents five perspectives, focusing particularly on the complexity of the relationship between municipal governmental officials and the national government not mentioning contributions from Fukushima Prefectural government officials. However, these “relationship challenges” tend to be easily expressed in society. Consequently, it may simply be that easily articulated grievances are surfacing. Furthermore, the paper does not present the underlying structural social context and administrative science perspective for the specific challenges discussed. Presenting analysis results without deeper exploration risks merely deepening divisions and hindering constructive progress. While resource constraints are emphasized, decontamination costs amount to approximately 4 trillion yen, representing a significant scale of response costs for an accident-causing environmental pollution compared to other cases. However, such comments may be perceived by the authors—who are psychologically attached to the research subjects—as an unethical attitude that fails to understand the suffering of the affected municipalities, as evidenced by the recorded testimonies. Therefore, it would be better to logically present the structure that inevitably leads local government employees to harbor dissatisfaction with the national government. Furthermore, while individualized responses tailored to personal psychological states may be effective in specific situations, this issue is fundamentally societal in nature, raising concerns that this perspective may be being overlooked.

Answer 1:

The reviewer’s comments are critical. First, regarding the perspective on “relationship challenges,” we agree that this should be explicitly acknowledged as a limitation of the study. Accordingly, we have added the following statement to the Limitation section (lines 653–661, pp. 28–29):

“Fourth, it should be acknowledged that the difficulties disclosed by the interviewees—such as their subjective distress or their strained relationships with the national government—represent only part of MGOs’ experiences and may not capture them in their entirety. It is possible that interviewees found it easier to speak about their difficult relationships with the government; if so, we must consider the possibility that some information remains undisclosed. There is an appropriate time to elicit and understand MGOs’ experiences. In this study, interviewees were able to share their stories because a long period had passed since the disaster, and conditions were in place to ensure that no individuals could be identified. However, building a longer-term relationship of trust between researchers and participants may create further opportunities for a richer and more in-depth qualitative inquiry.”

Second, the reviewer’s concern about the insufficient incorporation of structural social context and administrative science perspectives is well-founded. However, it is practically impossible for a single paper to comprehensively cover the diverse expertise involved in psychological, structural–sociological, and administrative science perspectives—and, if extended further, potentially political science, organisational science, and behavioural science as well. Importantly, although we are particularly interested in the psychological aspects of MGOs and in providing support for them, this does not mean we underestimate the value of structural, social-contextual, or administrative science approaches.

At the same time, our primary aim in this study was to analyse MGOs’ experiences from the standpoint of occupational mental health and psychological support, and to contribute findings that have practical relevance for frontline practice. We view the structural, social-contextual, and administrative science perspectives, as well as the psychological and occupational health perspectives, not as competing but as mutually important. Nevertheless, there are inherent limitations to accommodating all of these domains within a single manuscript.

In light of these considerations, and in order to clarify our position, we have added the following statements to the manuscript (lines 578–584, p. 25; lines 640–644, p. 28; lines 689–692, p. 30):

“While such individual-level mental health interven

---

## [Decision Letter · Decision Letter 1]

16 Dec 2025

Dear Dr. Hidaka,

Thank you for submitting your manuscript to PLOS ONE. After careful consideration, we feel that it has merit but does not fully meet PLOS ONE’s publication criteria as it currently stands. Therefore, we invite you to submit a revised version of the manuscript that addresses the points raised during the review process.

We look forward to receiving your revised manuscript.

Kind regards,

Sakae Kinase, Ph.D.

Academic Editor

PLOS One

Journal Requirements:

Additional Editor Comments (if provided):

This revised paper has been carefully considered by two referees. The referee's comment is attached. The authors should carefully consider the comment and resubmit, as soon as possible, an amended version of the paper.

Reviewers' comments:

Reviewer's Responses to Questions

**Comments to the Author**

Reviewer #1: All comments have been addressed

Reviewer #2: (No Response)

2. Is the manuscript technically sound, and do the data support the conclusions?

Reviewer #1: Yes

Reviewer #2: Partly

3. Has the statistical analysis been performed appropriately and rigorously?

Reviewer #1: Yes

Reviewer #2: N/A

4. Have the authors made all data underlying the findings in their manuscript fully available?

Reviewer #1: No

Reviewer #2: Yes

5. Is the manuscript presented in an intelligible fashion and written in standard English?

Reviewer #1: Yes

Reviewer #2: Yes

Reviewer #1: (No Response)

Reviewer #2: Please see the attachment.

1. General comments

The reviewer commends the authors' efforts to enhance the quality of research papers aimed at reducing the psychological burden on local government officials. High-quality qualitative research, unlike many quantitative studies, will bring immeasurable insights to humanity.

There may be value in achieving catharsis by addressing dissatisfaction with national government officials, which local government employees find easier to voice. However, from the perspective of solving fundamental issues, improving the relationship between local government officials and national government officials is essential. Without such efforts, as the authors themselves acknowledge, it risks exacerbating divisions. Of course, a single paper cannot resolve all issues concerning overly complex challenges as authors explained, but isn't the framing of the problem important? This is an unfair hypothetical question, and authors would be wise to ignore such abusive behavior from the reviewer. However, the reviewer became curious about what insights might be gained if these authors were to establish connections with national government officials and conduct a similar investigation.

Given that the authors have built relationships with research collaborators over many years, they should highlight the benefits this has brought to local government workers.

Authors may share this reviewer's comments with the local government.

.

Reviewer #1: No

Reviewer #2: No

---

## [Author Response · Author response to Decision Letter 2]

26 Jan 2026

We wish to express our appreciation to the reviewers for their insightful comments on our paper. The comments have helped us significantly improve the paper.

Here, we put our point-to-point responses to the reviewer’s comment.

The revised texts were marked by yellow in the manuscript.

[Reviewer 2]

1. General comments

> Of course, a single paper cannot resolve all issues concerning overly complex challenges as > authors explained, but isn't the framing of the problem important?... Given that the authors

> have built relationships with research collaborators over many years, they should highlight

> the benefits this has brought to local government workers. Authors may share this reviewer's

> comments with the local government.”

Answer:

We sincerely thank the reviewer for these thoughtful and encouraging comments regarding the value of qualitative research and the broader challenges surrounding relationships between municipal and national government officials. We agree that improving intergovernmental relations is a critical step toward addressing fundamental challenges in post-disaster reconstruction.

At the same time, the present study was framed to focus on the perspectives and experiences of municipal government officials (MGOs) to illuminate an underexplored experiential dimension of their engagement in coordinating with the national government during a decontamination project. While future research incorporating the perspectives of national government officials would be valuable, this was beyond the scope of the current study. We appreciate the reviewer’s reflections and take them as an important reminder of the broader institutional context in which our findings are situated.

In addition, although this lies beyond the scope of the present manuscript, our broader research project is ongoing, and we are preparing to share relevant findings with local governments. Through such feedback, we hope to enhance the practical and developmental value of our work and, modestly, contribute to both post-disaster reconstruction efforts in Fukushima and the health and well-being of the MGOs involved.

Q1:

> A) Line 49: “play key roles”

> The reviewer believes that the burden on city employees regarding decontamination

> projects stems from their relationship with residents, but t reviewer respects the authors'

> choice not to agree with this (primary group of interest in an academic paper should be

> focused). However, the reviewer has doubts about the social nature of the argument

> presented in this paper (whether it conveniently focuses only on easily discussable

> elements).

Answer:

Thank you for this thoughtful comment. We acknowledge the reviewer’s view that the burden experienced by MGOs in decontamination projects is closely tied to their relationships with residents, and we agree that this dimension is socially important.

At the same time, as the reviewer notes, the present study intentionally focused on a specific relational domain—namely, the relationship between MGOs and the national government—in order to examine an underexplored aspect of policy implementation from the perspective of MGOs. This focus does not imply that other relationships, including those with residents, are less significant. Rather, our analytical choice reflects the need to delimit the primary group and relational context of interest in an academic study.

We also take seriously the reviewer’s concern regarding the social nature of the argument and the possibility that the analysis may privilege more easily discussable elements. In response, we have strengthened the Limitations section to explicitly acknowledge that the experiences reported in this study represent only part of MGOs’ broader social experiences and may not capture their entirety (underlined/marked, lines: 676-678, p.29):

“Fourth, it should be acknowledged that the difficulties disclosed by the interviewees—such as their subjective distress or their strained relationships with the national government—represent only part of MGOs’ experiences and may not capture them in their entirety. It is possible that interviewees found it easier to speak about their difficult relationships with the government; if so, we must consider the possibility that some information remains undisclosed. MGOs’ experiences are embedded in a broader set of social relationships—including those with residents, colleagues, and other relevant actors—which were not fully examined in the present analysis. There is an appropriate time to elicit and understand MGOs’ experiences. In this study, interviewees were able to share their stories because a long period had passed since the disaster, and conditions were in place to ensure that no individuals could be identified. However, building a longer-term relationship of trust between researchers and participants may create further opportunities for a richer and more in-depth qualitative inquiry.”

Q2:

> B) Line 60: “mandated by the national government”

> The reviewer respects the authors' judgment that discussing this

> point is redundant, but considers it factually incorrect in the field

> of public administration from a legal perspective as well. Note that

> the reviewer does not state that this discussion should be conducted.

Answer:

Thank you for this important clarification. We agree that the phrase “mandated by the national government” may be legally imprecise from the perspective of public administration. We have therefore revised the wording to avoid this implication and to reflect the institutional context more accurately (lines 62-63, p.3):

“such as risk communication with residents under national policies and guidelines”

Q3:

> C) Line 65: “little is known”

> The validity of the analytical approach itself, which focuses

> specifically on the perspective of relationships with national

> government officials, is being questioned. Regarding the

> examination of the burden on local government employees, data

> on resignations and leaves of absence among young employees

> in each municipality may also be available.

Answer:

Thank you for this important comment. We agree that administrative data, such as resignation and leave-of-absence records, may serve as useful indicators of occupational burden among MGOs. However, such data do not capture how MGOs subjectively experienced and interpreted their distress, nor how specific aspects of intergovernmental interactions were perceived and meaningfully linked to their emotional burden.

To address this point, we have revised the manuscript to clarify that our study does not claim a lack of knowledge regarding the presence of occupational burden per se, but rather addresses a gap in understanding the subjective and relational dimensions of MGOs’ experiences, particularly in relation to interactions with national government policies and officials. The relevant passages have been revised accordingly (underlined/marked, lines 68-73, p.3-4):

“However, little is known about how MGOs subjectively experienced and interpreted such strained relationships in the context of post-FDNPP accident reconstruction efforts. In particular, it remains unclear which specific aspects of national government policies or officials’ behaviour MGOs perceived as contributing to their subjective distress. While administrative indicators such as resignations or leaves of absence may signal occupational strain, they do not illuminate how that distress was experienced or meaningfully linked to intergovernmental interactions. An in-depth examination of this intergovernmental interface from the viewpoint of MGOs is therefore essential for understanding how policy implementation processes shape occupational experiences, institutional trust, and barriers to reconstruction following a nuclear disaster.”

Q4:

> D) Line 77: “These characteristics may be the reasons for the

> reluctance of the affected MGOs to disclose their views regarding

> the national government”

> Is the authors' inference that local governments would not have

> agreed to the survey otherwise shared with the municipalities?

> Were deceptive techniques used in requesting cooperation for

> the survey from local governments?

Answer:

Thank you for raising this important ethical point. We want to clarify that this study was not conducted through municipalities as organisations, nor did it involve deceptive practices in requesting cooperation. As shown in the Methods section, two of the authors were engaged with the municipality through their professional roles in providing health support to MGOs, and the municipality was aware that research activities could be conducted as an extension of this work.

Within this context, participation in the study was sought from MGOs on an individual basis, and all interviews were conducted with the participants’ voluntary agreement and informed consent. No institutional pressure was applied, and participation did not depend on formal organisational authorisation by the municipality.

Q5:

> E) Line 79: “their anonymity must be protected;”

> The reviewer emphasizes the importance of respecting the

> autonomy of MGOs who contribute to the research. However, do

> the authors' concerns that this approach could potentially cause

> problems align with the perspectives of the MGOs themselves?

Answer:

Thank you for this thoughtful comment. We want to clarify that the approach to protecting anonymity and the timing of publication were closely aligned with the perspectives and decisions of the MGOs themselves, rather than being based solely on the authors’ concerns.

At the time of the interviews, some participants explicitly expressed that the public disclosure of certain narratives would not be appropriate at that stage, whereas they were willing to share their experiences. The authors maintained ongoing professional relationships with the MGOs after the interviews, and the decision to publish the narratives was revisited through discussion with the MGOs over time. Ultimately, publication proceeded only after the MGOs themselves judged that disclosure was appropriate and gave their consent.

In this way, the protection of anonymity and the delayed publication were implemented as part of a process that respected and upheld the participants’ autonomy. We have revised the manuscript to clarify this point (underlined/marked, lines 85-90, p.4):

“Therefore, when publishing the narratives of MGOs, their anonymity must be protected. In some cases, it is recognised that immediate public disclosure of such narratives may be inappropriate. Accordingly, publication may be delayed until the MGOs themselves judge that disclosure is acceptable, such as after transfers or retirements that reduce the risk of identification. Through such publication strategies, both the protection of MGOs’ anonymity and respect for their autonomy in research participation can be ensured.”

Q6:

> H) Line 89: “hypothetical model”

> Readers are interested in general hypotheses and how those

> hypotheses/research questions were verified. Can the insights

> gained be applied to improve operations on the ground?

Answer:

Thank you for this comment. We agree that readers are interested in the broader implications of the model. We want to clarify that the model developed in this study is exploratory and hypothesis-generating, rather than intended for hypothesis testing.

In response to the reviewer’s suggestion, we have revised the manuscript to clarify the model’s conceptual positioning and expanded the discussion to address how the model’s insights may inform practical improvements in on-the-ground operations, including support for MGOs and intergovernmental coordination processes.

At first, the authors have changed the term “hypothetical model” to “exploratory conceptual model” to clarify the hypothesis-generating nature of our study throughout the manuscript.

Second, we have added the descriptions regarding the role of our model to the first paragraph of “Exploratory conceptual model of MGOs’ subjective distress: Reorganising theories” subsection in the Discussion section (underlined/marked, lines 537-541, p.23):

“Here, we present the exploratory conceptual model and explain it. Note that this conceptual model developed in this study does not aim to test predefined hypotheses but to illuminate key points at which subjective distress emerges in MGOs’ work processes.”

Third, we have revised our manuscript to explicitly distinguish the explanation of our model from its application/implications for the ground. In “Exploratory conceptual model of MGOs’ subjective distress: Reorganising theories” subsection in the Discussion section, the following revisions have been added:

“...as well as those between national government officials and MGOs. Our model suggests that in the early phase of the post-radiation-disaster decontamination project, improvements in inter-ministerial communication, as well as communication between ministries and municipal governments, would likely play an important role in alleviating the mental burden on MGOs.” (underlined/marked, lines 570, p.25)

“Our model may be applicable for the support of the MGOs who were in a dilemma. While such individual-level mental health interventions are indispensable for supporting MGOs, it is important to recognise the risk that organisations may overly attribute the dilemma to “individual adaptability,” thereby overlooking conflicts rooted in structural deficiencies or flawed policy design...” (underlined/marked, lines 591, p.26)

“In Stage 3, the loss of residents’ trust in MGOs—and the resulting decline in morale and motivation—is described. Even when MGOs strive to fulfil their responsibilities while navigating the role conflict outlined in Stage 2, such efforts do not necessarily lead to positive outcomes; instead, they may result in a weakening of the ties between MGOs and residents. Our model suggests the need for an approach to improve morale and motivation among MGOs, which are influenced by exacerbated relationships with residents. From the perspective of public service motivation theory [29]...” (underlined/marked, lines 603-605, p. 26)

“Stage 4 represents the final phase, in which MGOs experience a sense of 'emptiness' as the core manifestation of their subjective distress. The implication of our model, which suggests that a sense of emptiness may be positioned at the final stage of the process of subjective distress among MGOs, is the importance of equipping mental health support with a diverse range of approaches to address this sense of emptiness. Because clinical outcomes of emptiness include suicide [35], the importance of addressing such emptiness is evident...” (underlined/marked, lines 624-627, p. 27)

Q7:

> 5. Methods

> A) How did authors explain the necessity of indicating gender?

Answer:

Thank you for this question. The indication of gender was not intended for comparative or gender-based analysis, but rather to provide contextual information that allows readers to assess the interpretability of the illustrative narratives, in line with the principles of thick description in qualitative research.

In the municipality where the participants worked, all MGOs who were directly involved in negotiations and coordination with the national government were male at the time. As such, indicating gender does not increase the risk of identification and does not conflict with our approach to protecting anonymity. We have clarified this rationale in the “Study design and recruitment of participants” subsection in the Methods section (underlined/marked, lines 120-124, p.6):

“local residents after the decontamination work started...Note that participants’ gender was reported to provide contextual information that enables readers to assess the interpretability of the narratives, consistent with principles of thick description in qualitative research. In the study context, all MGOs involved in coordination with the national government were male, and thus reporting gender did not increase the risk of identification or compromise participant anonymity.”

Q8:

> B) Line 243: "Line 104"

> Is the authors' judgment that this study does not constitute "insuffici

---

## [Decision Letter · Decision Letter 2]

22 Mar 2026

Municipal officials’ subjective distress in coordinating with the national government during the decontamination project of radioactive materials in Fukushima: A qualitative study

PONE-D-25-47633R2

Dear Dr. Hidaka,

We’re pleased to inform you that your manuscript has been judged scientifically suitable for publication and will be formally accepted for publication once it meets all outstanding technical requirements.

Kind regards,

Sakae Kinase, Ph.D.

Academic Editor

PLOS One

Additional Editor Comments (optional):

I have much pleasure in recommending this paper for publication. The manuscript has been substantially with changes according to reviewers' comments.

Reviewers' comments:

Reviewer's Responses to Questions

**Comments to the Author**

Reviewer #3: All comments have been addressed

2. Is the manuscript technically sound, and do the data support the conclusions?

Reviewer #3: Yes

3. Has the statistical analysis been performed appropriately and rigorously?

Reviewer #3: N/A

4. Have the authors made all data underlying the findings in their manuscript fully available?

Reviewer #3: Yes

5. Is the manuscript presented in an intelligible fashion and written in standard English?

Reviewer #3: Yes

Reviewer #3: I have read the most recent version of the article, and I have gone through the replies to previous comments.

I consider this manuscript interesting also for a non-Japanese reader, and I find it sheds light on the importance of "culture" in moments of crisis, even within administrative work as the coordination of decontamination efforts (which may seem a technical/"objective" kind of work).

I also found interesting that the tension between increasing expectations from the ministries, local administration and citizens resulted in such a distress for MGOs, which eventually culminated in a severe lack of meaning and pride, in some extreme cases loss of meaning for their own personal life. These considerations, although possibly not unique to the Japanese Society, can be important for future emergency management situations, also in other Countries, and this article is therefore quite interesting to stir a debate on those topics.

Moreover, I found interesting that the national dose limits policy changes contributed to the erosion of trust between MGOs and the local population, and did not help in the management of the transition towards the existing exposure situation. This aspect could be explored in subsequent studies.

Thank you.

.

Reviewer #3: No

---

## [Editor Report · Acceptance letter]

PONE-D-25-47633R2

PLOS One

Dear Dr. Hidaka,

I'm pleased to inform you that your manuscript has been deemed suitable for publication in PLOS One. Congratulations! Your manuscript is now being handed over to our production team.

Kind regards,

on behalf of

Professor Sakae Kinase

Academic Editor

PLOS One